# The Therapeutic Aspects of Embroidery in Art Therapy from the Perspective of Adolescent Girls in a Post-Hospitalization Boarding School

**DOI:** 10.3390/children10061084

**Published:** 2023-06-20

**Authors:** Nurit Wolk, Michal Bat Or

**Affiliations:** 1School of Creative Arts Therapies, University of Haifa, Haifa 3498838, Israel; 2Emili Sagol Creative Arts Therapies Research Center, University of Haifa, Haifa 3498838, Israel

**Keywords:** embroidery, art therapy, open studio, YPAR, adolescents, mental health

## Abstract

This phenomenological qualitative study explored the experiences of adolescent girls with emotional disorders from post-hospitalization boarding schools who embroidered in an art therapy open studio group. A Youth Participatory Action Research approach and the bioecological model were used to shed light on the therapeutic process of embroidery. Through a thematic analysis of the focus groups and interviews with 13 participants, we identified five themes specifically related to embroidery: (a) control versus release/freedom; (b) calmness that comes from the repetitive action and focus; (c) the experience of being exceptional versus conventional; (d) the “stitch through time” experience, which involves a dialogue with the past, present, and future through embroidery; and (e) the overt-latent layers of consciousness. The findings suggest that embroidery has therapeutic benefits for this population and supports psychological development. This study reveals that embroidery, whose threads are intricately embedded in society and culture, and may provide a unique and meaningful activity for young people in post-hospitalization boarding schools and enables a social and cultural exploration of self and community. Limitations of this study and recommendations for further research are also discussed.

## 1. Introduction

### 1.1. Embroidery

Embroidery is an ancient art form that falls under the umbrella of the textile arts, which stemmed from the ancient human need to dress, cover, and protect people and objects [1,2]. In contrast to textile crafts such as weaving or knitting that produce a surface, embroidery is carried out on an existing surface by inserting threads using a needle. Moreover, it does not fulfill a practical need but rather is intended more for beauty and decoration. It is like painting on a canvas, with the needle and threads taking the place of the brush and paints [3]. Throughout history, western consciousness has perceived embroidery through a gender-oriented lens, hence reducing its significance. Embroidery is perceived as a submissive, voiceless, and irrelevant female craft through a partial, patriarchal narrative that distorts gendered socio-cultural narratives [4,5,6]. The feminist movement has emerged to restore its forgotten gender-oriented history, such as its use by the suffragists in their 1910 protest [7], as well as the Bloomsbury Society’s advocacy of embroidery as a means of integrating women’s and homosexuals’ voices [8]. The connection between embroidery and the status of women is also evident in the lack of documentation of embroidery and the omission of women as creators [9]. As such, it lies on the seam line between the arts and the crafts and, as with other traditional arts, has not received enough attention in the field of art therapy [10,11].

### 1.2. Embroidery in Art Therapy

In recent years, the literature regarding art history, anthropology, and art therapy has reflected a resurged interest in the textile arts, especially embroidery. These media have gained recognition because of their valuable contribution to the illumination of individual and community social narratives as well as national narratives [12,13,14] in therapeutic work with women [1,14,15], soldiers (especially who have experienced trauma and individuals who cope with more general forms of trauma, as they allow distraction and the finished product creates a sense of value) [16,17,18], refugees [19,20], and individuals dealing with mental illness [21,22]. Akin to other textile arts, embroidery evokes memories, sparks playfulness, and allows for engagement and contact with the material, as well as the verbal content that arises in conversation during creation [23]. The introverted, feminine features of embroidery have been found to be significant in the creation of “emotional communality”—a term Rosenwein [24] coined to describe the communal space of consciousness that is created through embroidery in domestic spaces. Embroidery allows for the expression of personal and interpersonal secrets and messages that may not be voiced openly in other contexts [25]. In addition, embroidery, which references history and culture, allows communities to express and process traumatic narratives [14,20,26,27,28,29]. Apart from this, it can serve the economic needs of communities excluded from the public agenda, such as Mampola women in South Africa, the Bedouin women in Israel’s Negev region, and Ethiopian-Israeli women from the Almaz initiative in Jaffa, Israel [14,29,30]. This aspect makes embroidery an art form that is suited to the community art-therapy approach.

Beyond this, art therapists found that the creative process of embroidery connects key mental aspects such as internal and external mental spaces, creates intergenerational connections (not only because of its roots in tradition but through the act of interweaving the threads), and embodies characterization of animus and anima. These archetypal elements, according to Jung, exist within the male and female psyche and are key to psychological growth. Women need to develop the animus—the masculine element within them—while men need to develop the anima—the feminine element in the male psyche [31,32]. The use of thread and fabric represents feminine aspects, while the use of the needle represents a tension between masculine and feminine elements [33,34]. In addition, the cloth can represent a skin sheath while the piercing and fusing action can express both the creation of a skin sheath and its vulnerability and fragility [35,36].

Furthermore, art therapists have noticed three central features pertaining to the therapeutic aspects of embroidery: repetition, structure (working according to a pattern), and simplicity. These qualities may facilitate states of ‘flow’ [37] or rejuvenation [38], which can help create a feeling of calmness or increase energy levels [39]. In addition, rhythm and a sense of reward characterize the creative process in embroidery [40] and have been revealed as essential factors in the treatment of children and adolescents who have experienced trauma [41]. Given embroidery’s social, cultural, and psychological dimensions, it can potentially constitute an artistic language and a means of expression in art therapy, specifically in relation to adolescents with complex emotional difficulties [42,43].

### 1.3. Mental Health Challenges Faced by Adolescent Girls: Understanding and Addressing the Issues

Adolescence is a vulnerable period of development marked by significant and rapid physical and cognitive changes; peer relationships are especially important. Developmental challenges in this transitional phase include the development of autonomy, self-identity, and a philosophy of life [44,45,46,47,48]. Some believe that in the face of these developmental challenges, adolescent girls may tend to erase or disconnect from their own needs in order to conform to the expectations of others [49]. These challenges can lead to a mental health crisis manifested by sudden low moods, increased anxiety, depression [50], body image problems and eating disorders [51], suicidal tendencies, self-harm [52], and mental illness [53]. In this stage of development, there is a significant increase in the prevalence of psychiatric disorders. According to the National Comorbidity Survey Replication, 75% of psychiatric illnesses appear before the age of 24 [54]. Unfortunately, there has been a recent increase in the number of adolescents experiencing mental health crises; among adolescent girls, in particular, there is a higher prevalence of affective disorders [55]. This trend may be influenced by the COVID-19 pandemic as well as cultural and technological developments [56,57,58,59]. In regard to the latter, adolescent girls may face unique social and cultural challenges, including the pressure to be both autonomous and dependent, and may struggle with difficulties regarding sensory and emotional regulation—typical facets of adolescence [60,61].

The recovery process for adolescents following a mental health crisis is similar to that of adults and involves improving adaptive skills, increasing hope and acceptance, positive thoughts about the future, and enabling personal recovery within a social context. However, there are some differences that stem from age-related factors, such as physiological changes, preoccupation with self-identity, and significant external influences such as friends, social media, and cultural expectations [62]. Notably, following initial treatment, which may take a considerable amount of time, adolescents require monitoring and support through follow-up care. Additionally, the family plays a crucial role in the recovery process [63].

### 1.4. Out-of-Home Art Therapy That Addresses Mental Health Challenges in Adolescents

The family plays a central role in the development of social competence, emotional regulation, and mental well-being in adolescents [64], but family members may not always be capable of providing emotional support to adolescents during a mental health crisis. In these cases, therapeutic intervention may involve separation from the family and out-of-home placement, such as psychiatric hospitalization and out-of-home settings in the form of therapeutic boarding schools [65]. These mental health programs offer support, crisis management, and treatment. However, the separation from home, the new rigid structure and strict rules, and the encounters with others who are going through similar experiences can increase the number of challenges that adolescents with a traumatic background have to cope with [66]. Moreover, the programs and even hospitalization itself can be traumatic [67,68]. Therefore, when providing mental health care to this vulnerable population, it is crucial to take these factors into consideration and tailor therapeutic approaches accordingly. Having said that, adolescents often resist traditional mental health care due to their natural inclination to reject adult or parental authority [69]. In these cases, art therapy may be a more effective therapeutic alternative [70,71], as it provides adolescents with a bridge by which they can connect their inner worlds and external expressions [72]. Art therapy is administered to vulnerable adolescents in boarding schools who experience interpersonal and intrapersonal communication challenges [73,74,75]. It provides an alternative means of expression that circumvents speech, which can often be perceived as threatening [76]. In line with the evolving understanding of mental health, there has been a significant shift in terminology at a global level. Recently, the United Nations Human Rights Council replaced the term ‘mental illness’ with ‘mental health conditions’ (MHC) [77]. This new terminology represents a transition from a medical approach to one focused on well-being and a shift in emphasis from pathology to recovery [78].

The open studio approach, in its essence, aligns with this progressive shift. It acknowledges the importance of holistic well-being and recovery rather than solely focusing on the medical aspect. It also considers vulnerable adolescents with a wide range of MHC (including trauma) and provides a unique approach that fosters a sense of community, safety, and personal expression [79,80]. By emphasizing the healthy aspects of the self and providing a safe space, the model allows adolescents (working alongside therapists) to express personal content while working around resistance and suspicion [79,80]. In addition, by combining the language of art and art materials in a communal setting, adolescents can move between personal and group spaces and regulate their emotions [74]. The current study was conducted in a post-hospitalization boarding school in central Israel, where many girls chose embroidery as a form of creation in the open studio space of the inpatient art therapy program. It aims to illuminate the potential therapeutic benefits of this art form by listening to these girls’ voices and documenting their experiences with embroidery.

Since embroidery incorporates personal, cultural, and social aspects, we chose to use the bioecological model in this study [81,82]. This theoretical framework perceives the treatment of mental illness as having a social-cultural context, and the communal art therapy approach is one expression of this view. Textile art, in its various forms, is a prominent example of the practical implementation of these ideas. The bioecological model is described as a series of circles that contain each other: (a) the microsystem—the individual; (b) the mesosystem—the environments with which a person interacts; (c) the macrosystem—the wider cultural and social environments (some also include the ecosystem, which has an indirect effect); and (d) the chronosystem, which refers to the aspect of time [82]. This model can provide a broad and comprehensive lens for observing the experience of embroidery among adolescent girls while emphasizing four principles relevant to the phenomenon presented in the study: 1. A person’s psychological development is not separate from his or her social and cultural environments; 2. There is interaction, movement, and transitivity between the environments, and this is essential to mental health; 3. Time, as a chronosystem, is another dimension that must be considered, especially in treatment processes that unfold over time; and 4. The individual also participates in proximal processes, which involve the reciprocal interaction with the immediate environment, including significant people, objects, and symbols. Using this theoretical framework and the Youth Participatory Action Research (YPAR) approach [83] involving focus groups, interviews, and the observation of embroidery artworks, the present study sought to answer two main research questions: (1) What is the adolescent’s subjective experience of embroidery in an open studio in a post-hospitalization setting? (2) How are the two-way circles of influence that encompass the embroiderer, the embroidery process, and the environment represented?

## 2. Method

### 2.1. The Research Approach

This study on the therapeutic benefits of embroidery as experienced by adolescent girls’ post-hospitalization takes into consideration the important perspectives of patients [84]. To gain a deeper understanding of the research question, a phenomenological YPAR method was used [85], which involved observing the expressions of the 13 adolescent female participants through multiple channels. The ultimate goal of this approach was to gain insights into the girls’ personal experiences of embroidery in the open studio setting and to gain a better understanding of the therapeutic benefits from their perspective. The phenomenological nature of the research allows for an in-depth exploration of the subjective experiences of the participants. This research method was chosen as it allows for joint investigation together with the participants and therefore can lead to empowerment. The study aimed to achieve its goal through a reflective circle, in which the participants collect and analyze the data and determine the next steps [86]. Additionally, the YPAR method addresses the power relations between therapists and participants and allows for a more comprehensive understanding of the research topic. It is important to note that this method also allows for the power to be shared between the researcher and the research subject during the discussion and conclusion of the study [87].

### 2.2. Participants

A total of 13 Israeli adolescent girls aged 15–18 (M = 17.42, SD = 1.164) participated in this study. They were students in a post-psychiatric hospitalization boarding-school that is known for its treatment of a wide range of mental disorders, including: depression, anxiety, eating disorders, personality disorders, accentuation of personality traits, PTSD and adjustment disorder, suicidal ideation, self-harm, ADHD, and learning disabilities. 

All participants had attended embroidery groups held at the boarding-school in an open studio setting for a period of at least 12 months to 2 years between 2020 and 2022. Participants were selected using criterion sampling [88].

### 2.3. The Research Process

After receiving ethical approval from the Ministry of Labor, Welfare and Social Services and the Ethics Committee of the Faculty of Social Welfare and Health Sciences at the University of Haifa, we invited the adolescents who participated in the embroidery sessions in the open studio to take part in the YPAR, which focused on the experiences of embroidery in reference to the embroidered pieces. In order to ensure the involvement of minors and address the vulnerability of this population, we obtained explicit consent from the parents or legal guardians for the participation of all 13 girls. This consent was obtained in addition to the ethical approval granted.

The first author, who is both an art therapist at an open studio boarding school (see Section 5) and a researcher, oversaw the YPAR project. To allow participants maximum freedom of expression, enable an open discussion during the research (the focus group and interviews), and minimize potential bias and promote unbiased results, the researcher avoided moderating the participants’ art-therapy embroidery sessions (see Section 2.3.1, which provides more details on the embroidery session in the open studio). She was replaced by another therapist who moderated sessions for Group A. For Group B, there was no change as it had been facilitated by two other art therapists from the beginning.

#### 2.3.1. The Embroidery Session at the Open Studio Post-Hospitalization Boarding School

The girls who came to the embroidery sessions came by choice; participation is not mandatory according to the open studio model [80]. The sessions were held once a week, with 4–7 participants in each session and moderated by two art therapists. In each session, a variety of embroidery materials were offered, and each girl was encouraged to embroider according to their personal choice. The girls and two art therapists sat around one table while the therapists took part in the conversation, embroidered alongside the girls, and responded with sensitivity to things that were said in the general conversation or in relation to the creative process. Each session started with a short ritual whereby the girls were invited to express themselves using thread and embroider freely on a shared cloth. After that, each participant worked on their own personal project. At the end of each session, a joint observation of the embroidery was conducted, in which the girls were invited to share their thoughts, feelings, and remarks.

### 2.4. The YPAR Procedure

The YPAR procedure involved the methodological triangulation [89] of two methods: in-depth interviews and focus groups that took place in two phases: The first phase consisted of a focus group discussion of 13 participants conducted in two groups (7 participants in Group A and 6 participants in Group B); the second phase consisted of personal interviews with 11 participants who agreed to be interviewed.

In the first phase, keeping the discussion as open as possible, the researcher presented the research question and asked the participants to refer to their embroidery experiences. While they embroidered, the participants chose topics that they would like to explore. To ensure flow and focus, the researcher wrote the topics proposed by the participants on a board. Each focus group meeting was recorded, and accompanying embroidery pieces were photographed. The focus groups lasted about an hour. The number of focus group meetings was determined by the participants—Group A held 4 meetings, while Group B held 2 meetings.

In the second phase, 11 in-depth interviews were conducted with participants from the embroidery sessions who had consented to be interviewed, which included observation of their artistic products. Participants were also able to relate to photographs of embroidery pieces of other girls who participated in the group. The aim of the interviews was to gain a more personal perspective and further understanding of the embroidery experience in a therapeutic context and to focus attention on issues that may not have received enough attention in the focus group.

The data collection phase involved conducting interviews until saturation was achieved, meaning that new insights relevant to the identified themes were no longer being generated [90,91,92]. The researchers determined the saturation point through ongoing analysis of the data based on the conclusion that further data collection would not significantly contribute to the theoretical understanding already established [93].

### 2.5. Data Analysis and Trustworthiness

The data consisted of transcriptions of the focus groups and interviews, which included references to the participants’ embroidery work; these focus groups and interviews allowed participants to reflect on their embroidery. Analysis of the interviews and embroidery was conducted according to the six steps of thematic analysis described by Braun and Clarke [94]. To start with, both authors went through all the transcripts on their own to gain familiarity with the data. Afterward, they searched the data for items related to their research question and categorized them based on their meaning. In the third phase, they analyzed the categories to identify common patterns among all participants. In the fourth and fifth phases, the researchers defined and labeled the patterns to capture their central idea and significance. Eventually, they created a visual representation of the patterns (a thematic map). While analyzing the data, the researchers maintained open discussions to note their personal beliefs and biases in order to prevent them from influencing their conclusions. They also ensured the consistency and reliability of their analysis by comparing their individual analyses, discussing discrepancies, and agreeing on the content and interpretation of themes [95].

The authors followed Gillian’s [96] third principle of critical visual methodology, which prioritizes participants’ explicit observations and interpretations of their embroideries over researchers’ opinions (see also [97,98]). The authors’ analysis integrated participants’ experiences regarding specific embroideries, as detailed in the findings below. The analysis was conducted in Hebrew, with the final report translated into English. We ensured the accuracy of the translation by confirming the meaning of the original text and consulting a bilingual language editor. Given that embroidery is not commonly used in art therapy sessions in Israel and even less so for the purpose of research, the sample had to be limited to the first author’s facilitated groups (see Section 5).

## 3. Findings

Following data analyses of interviews, focus groups, and the observation of embroidery pieces together with the participants, five themes unique to embroidery and to this population of adolescent girls were identified: (a) control versus release/freedom; (b) calmness that comes from the repetitive action and focus; (c) the experience of being “exceptional versus conventional; (d) the “stitch through time” experience, which involves a dialogue with the past, present, and future through embroidery; and (e) the overt-latent layers of consciousness. Each of these themes references the three circles of bioecological theory: the personal embroidery experience in relation to self (micro), in relation to the embroidery group (mesosystemic), and in relation to society (macro). The fourth theme, “stitch in time,” encapsulates the notion of the fourth circle, which relates to time (chrono). Movement from one circle to another was identified as well. Table 1 summarizes the associations between the themes and the bioecological circles.

### 3.1. The Dialectic between Control and Release/Freedom: Use of Patterns as Opposed to Freestyle Embroidery

The participants’ descriptions of their experiences reflect a dialectic between control and the release of control or freedom in three relational circles (the micro, the meso, and the macro). Participants described how they were able to shift from embroidering using a pattern to freestyle embroidery. In most cases, the participants reported that their embroidery during the period in which they were hospitalized mostly entailed the use of patterns. After leaving the hospital, they turned to freestyle embroidery. The girls reflected on the question of which style of embroidery—embroidery using a pattern or freestyle techniques—was the most effective in helping them deal with their mental challenges.

When reflecting on her cat embroidery during her interview, Participant 1 explained the growing freedom she felt in relation to her embroidery work as a developmental process in the micro circle. In her cat embroidery (Figure 1), one can see the freedom she expressed when she broke the “rules” and added spots of blue acrylic paint.

“The liberty I took when working with the cat is a recent development. Two years ago, when I started embroidering in the hospital, it took a long time before I even agreed to try. At first, I mostly used patterns for my projects. Later, I began choosing patterns for inspiration and sometimes ventured to create my designs. Ultimately, I started doing what I wanted. It was a process to get to this level of liberty.”

A few participants (*n* = 3) were ambivalent about the freestyle embroidery options available to them during art therapy sessions at the boarding school. These individuals reported that they struggled with the idea of moving away from embroidery involving the use of patterns as it had helped them cope during their time in hospital. In her interview, Participant 10 expressed frustration with her attempts to cross-stitch (See Figure 2a,b, cross-stitched tree), stating that she felt that she has been unsuccessful because she has not been provided with appropriate patterns.

“I believe there’s a significant difference between cross stitching and freestyle embroidery. When doing cross-stitching, there are clear instructions on where to insert the needle and how to pull the thread. It can be easier in some ways because you know what you’re supposed to do. In the hospital, kids follow examples in books or use fabric with printed cross-stitch patterns, which makes it easier because then there are clear guidelines. But freestyle embroidery is more open-ended and there’s no set pattern to follow. For example, when I was working on the tree leaves, there were colorful flowers on the tree and it was difficult to know where to put them. The art therapists didn’t provide clear instructions and that was discouraging.”

The topic of embroideries made with a pattern as opposed to freestyle embroidery was mentioned frequently as the participants reflected on the embroidered stitches and elements of particular embroidered pieces, such as cross-stitches versus freestyle stitches, the size of the stitches, and the themes they chose to depict. Both in the focus groups and in the interview, Participant 7 commented on Participant 2’s embroidery, explaining that her messy stitches reflect her erratic menstrual cycle.

“Listen, menstruation is not something that is neat and organized, is it? So obviously the stitches there on the pad that she sewed are not neat and organized either.”

The mesosystemic circle was expressed through the girls’ reports of experiencing a sense of freedom due to the fact that the group was open to non-structured tasks, i.e., embroidery without a pattern, as well as it being supportive and accepting. They credited this positive environment to the non-judgmental atmosphere and the fact that the therapists embroidered alongside them, creating a feeling of equality. In her interview, Participant 6 emphasized the fact that the group setting allowed her to express herself freely and break away from traditional embroidery patterns. She described how she embroidered a text upside down (Figure 3), but instead of feeling as though she made a mistake, she felt a sense of release.

“You can express everything here {in the group-W.N}, like embroidering on a pad, on a floor rag […]. It was really cool, like it’s not something normal, so to speak. As if you can do with it (embroidery) what you want. We had this activity of making each a small square. I just scribbled on mine. And I also wrote the “meow” {A cat’s mew from a children’s TV show-W.N} upside down, oh my, it’s terrible. And it is, for example, very inaccurate. But I had fun doing it too, it was wild to do such a thing. It was the opposite of doing the correct thing, I just went with the flow. It’s wild like that and it was cool.”

### 3.2. Calming—Repetitive, Focusing Action

The theme of embroidery as a calming activity emerged as a significant topic in the microsystemic circle, where participants divulged the experiences they had and the feelings they felt during the creative process. They believed that a feeling of calmness can be achieved through the repetitive movements involved in embroidery, the tactile quality of the embroidery on the fabric, and the sound of the needle piercing the fabric, as well as the need to concentrate.

“It’s very relaxing. {embroidery-W.N} Both the repeated action and focusing on one thing for a long time is very relaxing, it’s pausing everything and lets one focus on one thing.”Participant 5 (interview)

“Physically, it’s something fun to touch and mess with, it’s soft and cozy, and it’s comforting, and its touch calms a lot of people… The “click” of the needle, the sound of the needle moving through the fabric, it relaxes me, like white noise, like the sound of a waterfall.”Participant 9 (interview)

Additionally, several girls expanded their perspectives to include the mesosystemic and macrosystemic domains. They explained that participation in the embroidery group brings with it a sense of calm and that remaining in the group contributes to this state of tranquility. They further argued that their need for calmness arises from the turbulent nature and demands of Western culture. For example, in the focus group, Participant 1 said that being in a group had a calming effect on her, and she also expanded her thoughts to include environmental, cultural, and social dimensions.

“I get into the zone, I am with myself, maybe it’s kind of pausing for a moment, from the day for something that is relaxing, some kind of situation with yourself… and we sit in a circle around the table, and it’s, like, it kind of focuses the group because everyone gets into some kind of similar mood…There are distractions in today’s culture, lots and lots of external things that interfere with our ability to take a second, to stop for a moment.”Participant 1 {While embroidering-W.N}

### 3.3. Being Exceptional versus Conventional

The theme of being exceptional as opposed to conventional was reflected in the three ecological systems. In the microsystemic circle, participants described this dimension as enabling the creation of unique and unusual embroidered pieces. In relation to the mesosystemic cycle, they explained how, as young adolescents, their engagement with embroidery was unconventional, and how their group was unique and different. In macrosystemic terms, they spoke about how they often felt like atypical adolescents, having departed from the conventional path instead of being “mainstream”. Participants also noted that embroidery may not be suitable for everyone, including those who participate in the open studio and prefer not to engage with its intricate and time-consuming aspects, such as the use of a pattern and the repetitive work that requires patience and time. Some people might find this unappealing or challenging. Participant 4 expressed the intrapersonal (micro) dimension during her interview by explaining that two of her embroidered pieces—a rhinoceros embroidered on a shimmering fabric (Figure 4a), which combines thread and unexpected materials, and a snail embroidered on a flowery background (Figure 4b)—provided her with a unique means of expression and allowed her to express the persistent feeling that she is different from the norm.

“I love animals and the combination of snails on a flowery, glittery background seems weird, but in a good way. Weird—nice because it’s the glitter fabric. It’s fun for me that it doesn’t connect. As if there is this game between a very specific thread and a very specific substrate of mine. It’s like everywhere I’ve been I’ve been a “different thread” in a “different substrate”, sometimes it connected well, sometimes it looked cool and sometimes it didn’t.”

Referring to the group, the meso- circle, Participant 6 highlighted the shared sentiment among the girls that the embroidery group is distinctive and unconventional compared to the general population due to the sensitivity and meticulousness of its members.

“It was a special group. There were, absolutely, girls who came here and they didn’t stop until they got something perfect, absolutely, it’s like that with a lot of people in general. But here it is something that preoccupies us, mainly because perhaps we come from a background that made us more sensitive. I think we are very, very sensitive girls. We exhibit more sensitivity than other people. The sensitivity is very noticeable. Because I pay attention to every very small detail, I am sensitive to everything. Every terribly small detail seems bigger to me than to other people. Like in embroidery.”

The girls viewed themselves as unconventional but also as special because they create through embroidery. The macrosystemic circle manifested itself in their comments on how embroidery is an unusual activity for their generation (hence contributing to why they are special). They also viewed the post-hospitalization boarding school as being similarly unusual. In the excerpt below, Participant 1 highlights the uniqueness of the embroidery group and quickly shifts the conversation to the boarding school, stating that it is unconventional itself in comparison to other boarding schools. In this excerpt, one can see how this theme of “uniqueness” connects to the theme of “patterns”.

“Because embroidering is “odd,” it is also the thing that makes it more special. Because it is valued. For example, it is rarer and more unique to see a greeting card in a store with embroidery on it, compared to a greeting card with a drawing on it. People see boarding schools as something (negative) different and strange, because most of the boarding schools you hear about are for teenagers who were kicked out of the house. This boarding school framework is unconventional.”

### 3.4. Stitch through Time—A Dialogue with the Past, Present, and Future

When they discussed their embroidery, the girls communicated a strong preoccupation with the time dimension, which encapsulates the chronosystemic circle as well as the microsystemic circle. The microsystemic circle refers to their personal experiences of time, such as the rate of progress of their embroidery, their past experiences, present situation, and future possibilities in relation to their work. Additionally, the chronosystemic circle manifested itself in their reflections on the historical context in which they were creating their embroidery, placing it within a broader framework of time.

An expression of the microsystemic circle can be seen in the words of Participant 5, who refers to embroidery as an experience in which one progress slowly—similar to psychotherapeutic treatment, where the progress seems to be slow but, in the end, the progress is there.

“I think that the repetitive and sisyphean process of embroidery is similar to those things we do repetitively where we may not always feel like we are making progress even though we are. I again compare it to the soul like going to therapy say once a week and again and again and again. And in the end, step by step, each treatment is like a stitch, in the end there is progress. There is progress at the end.”

Half of the girls mentioned the fact that they are engaging in an activity that has existed for a long time. Participant 7’s words reflect a dialogue between the microsystemic circle, her personal identity as a 17-year-old girl, and macrosystemic circles that also touch on chronosystemic meaning, ancient culture, or ancient times:

“Embroidery is an activity that belongs to a time from many years ago, women used to do it. I think it’s worthy of appreciation, that I’m 17 and I embroider, wow, nice, I want to see more 17-year-old girls embroider. I don’t think it’s something that is commonly done in our times, today. There is no 17 years old girl who suddenly wakes up in the morning and says “Wow, I want to embroider.”

In addition, most of the girls talked about how embroidery allowed them to take a “break” from other things. Half of them viewed the break in mesosystemic terms, recognizing that the group setting provided a designated time for them to pause and spend time together. Most of them describe the present era as characterized by information overload. As Participant 11 put it, embroidery offers a brief escape from the constant barrage of information and stress in today’s world.

“You know that you have an hour like that, and you know that it {the embroidery W.N} relaxes you, and it does you good, it is good to take this break in the day. It’s such a pause […] and then concentrating on one thing. Chasing after money… I’m talking about life that isn’t really related to the boarding school. I’m talking about life in general: The pressure, the school system, feeling busy, pressure, the responsibility on children, family. Sometimes you need to breathe. There are many people who need this break. Maybe that’s (embroidery) what helps you to calm down, that focuses you that allows you to be centered.”

During the research, as the girls discussed their embroidery work, they also shared their thoughts and concerns about the future. They wondered whether they would have the opportunity to continue embroidering in future programs and whether they would have enough quiet time for themselves as they got older. Additionally, they expressed curiosity about what would become of the research findings. In fact, they affectionately gave the main researcher the nickname “Nurit of the Future,” joking about how her recorded words might be used as evidence in the future. During her interview, Participant 1 joked: 

“And now you’ve stopped for commercials, wait—we’ll immediately return to the podcast of “Nurit of the Future”, Nurit, when you’ll edit it and listen to it, good luck, good luck with your research in embroidery… I left you a message.”

### 3.5. The Overt-Latent Layers of Consciousness 

This theme brings together three forms of expression: the hidden versus the explicit, the personal versus the social, and the unspoken versus the voiced. The first type of expression, which is more microsystemic in nature, can be seen in the way the girls refer to hidden as opposed to visible aspects in the substrate, stitches, and in the knots. In her interview, Participant 5 speaks about the process of untangling knots on the back of her embroidery, which has given her insight into her experience with depression. She explains that the back side of the embroidery represents her inner self, while the front represents the expectations of society that are not consistent with her true feelings. She reflects on her realization that examining the knots on the backside of her embroidery offered a metaphor for her challenges, leading her to take a closer look at her struggles. Participant 5 shared her insights relating to the practice of focusing on the knots in her embroidery as representing the attention she devotes to her difficulties.

“Here I think I looked behind to see knots that I think were really my biggest problem—the knots from behind. I wouldn’t have noticed, and it would have gotten completely complicated, and it was impossible to solve it because it’s a crazy knot. It’s like the thing in front, everything looks perfect and in order, but you don’t see the back, you don’t see what’s going on inside the person, and sometimes you don’t understand why it (the embroidery) isn’t finished. But (this is exactly the problem) there are ties behind, and you don’t see it. But over time I embroidered more slowly, I looked at the back more often, I just learned to look back and see that everything is fine from time to time.”

In one focus group session, Participant 11 spoke about how, through the back side of the embroideries, the emotions that the embroiderers have experienced can be revealed:

“You can really see and feel the parts where we got angry, and the parts where we were like “Oh man, no no” {imitating someone who is dissatisfied and making the group laugh W.N} […] On the one hand, you can see the beauty and the investment, and on the other hand, you can see the irritation and the frustration. It’s seeing what’s behind the scenes, let’s say that in an acrylic painting, it’s very hard to see because everything is in layers, and you can’t see the back.”

A second type of expression, which is macrosystemic in nature, reflects the tension between the intimate and the personal—the often-concealed internalization and the externalized social protest or statement. Participant 3 refers to her embroidered text cynically (Figure 5) and perhaps somewhat critically, saying that, although she usually creates art in a very personal way, she also needed to make her voice heard publicly. In her interview, she says:

“This text talks about once I’ll be mentally stable then ‘watch out’ I’m going to be the best (Some day in the future). This statement is more with myself, most of the things I create are just me with myself, but here maybe because I’ve been becoming more stable recently, I see that I’m gaining self-confidence, and I’m more myself, so when I say such a statement in embroidery, I also direct it to the environment. “It’s all for you”, for society in general. it’s like ‘watch out’.”

The overt–latent theme was eloquently conveyed by the participants when they discussed how embroidered pieces are the perfect gift, as they can contain and encapsulate personal elements such as handwriting, content, effort, and intention (microsystemic features). They also mentioned the versatility of embroidered pieces, i.e., the fact that they can be gifted to others or put on display for all to appreciate (meso-macrosystemic features). Below are excerpts from one of the focus groups.

“Embroidery is good as a gift because it is handmade.”Participant 8

“Because you can create a lot of things with it, and it’s very beautiful, and the fact that it’s handmade shows that someone made an effort and made something special for someone else.”Participant 10

“You can really see as if from the back too, you can see the process. In fact, […] You can see the effort. I think you can feel it. So that’s what makes it such a meaningful gift. On the one hand you can see the beauty and the investment, and on the other hand you can see the irritation (laughing) and the frustration.” Participant 11

A third type of expression, which has mesosystemic qualities, was conveyed by the girls when they explained that there is a difference between what they can say in other forums (overt) and what they can say in the embroidery circle (latent). In one focus group meeting, Participant 11 described how the embroidery group allows her to speak her mind and allows the girls to open up about their daily experiences and struggles, as well as offer advice and support to one another.

“In an embroidery group there is more intimate conversation, maybe because the act of embroidery is repetitive, and you don’t have to think all the time, and maybe because we are just girls. Here I feel that it is possible to open up and say what wasn’t good today, who annoyed me and who didn’t, which instructor annoyed me, which kid annoyed me, or vice versa. […] In other groups, I won’t say things like that, unless I feel comfortable. Most of the times in the other groups they are more like straightforward, learning and all the classes there are, like, I’m also in social psychology, philosophy and photography and there it’s not like that, it’s not intimate. […] Here we also advise each other. And it happens a lot. Giving advice in personal life as well, someone says, let’s say something happened, and also regarding the works themselves, if I wondered whether to do one thing or another in the embroidery and the girls helped me and it was like I went with it.”

Regarding an embroidered piece depicting menstruation that was made on a menstrual pad (Figure 6a–c), Participant 2 explains how a topic such as menstrual pain, which is not discussed publicly and, when discussed, is a topic that arouses negative reactions, can be openly discussed in the embroidery group. She explained that the apparently blunt words do not make her any less feminine or delicate, like the embroidery itself.

“In the dorms, I once shouted that I was in pain because of my period, and they told me it was not appropriate […] In the embroidery group I started embroidering because it (menstruation) hurt me, really hurt me. Other girls identified with it. I knew there was something beyond that. It’s not just me who suffers from my period and it’s not just me who will connect to that.

Through embroidery, I can convey that I am in pain and I have my period and I, it’s difficult and yes, it’s something more feminist and it’s okay to talk about it, about menstrual pain and menstruation in general also in such a way, through embroidery, that is delicate and feminine, precise and sketched and [made using] such delicate motor skills. It is also possible to be a feminist in this. It doesn’t have to be masculine and rude and blunt; it can be strong and blunt anyway.”

## 4. Discussion

This YPAR study sought to shed further light on the therapeutic aspects of embroidery from the perspective of adolescent girls living in post-hospitalization boarding schools. These girls referred to their experiences of embroidery in open-studio art therapy sessions, focusing on the following questions: “What is the ‘lived experience’ of embroidery?” and “What are the inter-influential circles expressed by the embroiderer, the embroidery, and the environment.” The study found five main themes that reflect the therapeutic aspects of embroidery, particularly in relation to adolescent girls. Embroidery stimulates shifts between controlled and freestyle techniques, inspires a sense of uniqueness and unconventionality, provokes a preoccupation with aspects of time, enables a subtle sense of interplay between introversion and extroversion, and provides a source of relaxation and tranquility.

Akin to other textile arts, embroidery, given its repetitive patterns and order, is considered a soothing or grounding activity [1,2,10]. However, this study provides a unique perspective on the structured aspect of embroidery. Specifically, the study found that the transition from structured to unstructured embroidery was an effective way of adapting the embroidery to the mental states of participants. While the adherence to embroidery patterns provided emotional support during hospitalization, participants preferred a freer style as they overcame their mental health challenges and achieved a more balanced state of mind. The girls found that embroidery is a versatile art form that allows for both structured work with patterns and more creative freestyle work. Therefore, creating a balance between structured (using a pattern) and unstructured (freestyle) embroidery can offer both emotional support and a sense of release, depending on the individual’s mental state. Conforming to an accepted pattern while simultaneously breaking free is a typical activity for adolescent girls, one which conforms with the developmental stage of puberty. Moreover, after hospitalization, the girls required a template that could contain and provide order in their state of mental chaos, but they also wanted to free themselves from the rigid patterns imposed during hospitalization [62,63]. This idea goes hand in hand with findings based on the perspectives of adolescents, which reveal that rigidness experienced during psychiatric hospitalization does not aid the recovery process [68]. This idea of ‘breaking the pattern’ is in line with the adolescent developmental challenge of pushing back against parental authority and rebelling against expectations of conformity and assimilation [48,99] while also seeking and needing parental support [64].

The idea that embroidery allows for a dialectic between adherence to patterns and freedom from patterns is associated with the feeling that embroidery promotes calm. This calm is characterized by a “flow” state of mind and long-term improvements in mood, which are generally observed in art therapy (particularly with respect to traditional art forms) and conceptualized by Collier and Von Károlyi [38] as “rejuvenation;” it is an appropriate response to the needs of this vulnerable population. The question arose as to which form of embroidery is more relaxing: embroidery according to a pattern, such as cross-stitching, or freestyle embroidery. Reflecting on this question helped some participants identify which style matched their internal feelings; however, for others, the need to choose between embroidery styles caused anxiety and agitation. This highlights the importance of maintaining a calm environment and should prompt therapists to consider the ways in which they can offer creative solutions to promote calmness.

The need for serenity among adolescent girls prompts further inquiry into their needs, particularly their need for recognition and visibility. This is closely related to the establishment of self-identity during adolescence [45], when adolescents (and especially adolescent girls [49]) are more likely to rely on recognition from their environment to shape their self-perceptions and achieve a sense of belonging [47]. Through references to their embroidery, the participants clarified their needs and indicated a desire for recognition as individuals with distinct qualities, such as uniqueness or extraordinariness, which we called the exceptional versus conventional theme. This desire was mentioned by participants in connection with their embroidered creations, their group, and their place in society. They felt that girls their age engaging in embroidery is unique, that the group was “special” and different from other therapeutic groups they knew, and that they, as individuals, did not conform to the norm.

Notably, while discussing their embroidery, the girls present the “different” and the “other” as special. This aspect may be consistent with the therapeutic process that allows the “unconventional” to become “unusual and special.” This perspective, similar to the art brut philosophy, which recognizes the different and the abnormal and repositions it as special and unique [100], is also reflected in the Japanese art of kintsugi. Through repairing broken ceramic vessels with gold, kintsugi exemplifies the idea of embracing imperfection and turning flaws into unique features [101,102]. The unique artistic expression of adolescent girls who faced a crisis and underwent hospitalization not only provides meaning but also reflects their self-identity in relation to the community and society. During adolescence, developing a positive self-identity can be a challenge, especially for those who have experienced significant difficulties [44]. It seems that embroidery allowed the participants to adopt an aesthetic approach, embracing their “differences” as something exceptional and promoting self-compassion and acceptance of their own unique traits [103]. This aspect is in line with current critical trends for the treatment of mentally challenged individuals, such as the recovery model and the neurodiversity movement that emphasizes empowerment [104], and highlights the unique strengths and abilities of individuals who cope with mental/cognitive challenges [105]. In addition, the transition, from the perception of oneself as “different” to the recognition of the self as “special” can be indicative of internalized therapeutic progress, resulting in a shift in self-perception. This particular attribute is one of the hallmark characteristics of post-traumatic growth [106] and has relevance to the research population. Adolescents who have been hospitalized for mental health issues often have a history of traumatic experiences [66]). In addition, the hospitalization process itself can be a complex event that leads to a feeling of vulnerability and loss of control [67]. In light of this, the “stitch in time” theme can also provide insight into the trauma recovery process. The girls’ experience with embroidery allowed them to contemplate the concept of time, interruptions, and continuations, which correspond to Bion’s concept of “caesura” [107]. This term describes the paradoxical concept of being both separate and continuous at the same time. The caesura’s therapeutic value lies in the ability to recognize the continuum and explore the interval, even in situations of trauma and fragmented life events. The embroidered line embodies both interruption and continuity, enabling individuals to engage with the contradictory spaces they encounter. As Homer [12] asserts, embroidery allows for the visualization of time. Creating an embroidered piece prompts participants to consider both linear aspects of time, such as measuring progress, and interrupted aspects of time, such as the possibility of jumping through time, contemplating the future, reflecting on the past, or starting a new project while leaving an unfinished one behind. Additionally, the concept of taking a “time out” for the self was emphasized by participants as an essential aspect of embroidery as a form of therapy. This notion, which was found by Potter [43] to be true regarding textile arts in general, is particularly relevant in the context of the fast-paced and evolving era in which we live. This also highlights the importance of identifying the therapeutic needs of vulnerable adolescents in today’s world. Thus, embroidery can provide people the opportunity to take a break from the chaos of daily life. The thread, which embodies the concept of time, also expresses the overt and latent. The fact that the stitches are visible in the embroidery allowed the participants to reflect on both the hidden and visible aspects of their self. This was particularly notable when the girls discussed mental difficulties that are not always apparent from the outside. The girls provided intrapersonal, group, and social-cultural interpretations of this theme. It is clear that observations of the stitching on both sides of the fabric facilitated a dialogue between the internal and external parts of the self. Furthermore, the group setting allowed for an open and intimate discourse that is often silenced, similar to the concept of “emotional communality” in the embroidery groups described by Rosenwein [24]. Discussions in the mesosystemic circle (the group) were often expanded to include broader socio-cultural themes, such as discrimination against women and feminism. This discourse was influenced by the archaic women’s circle represented in the macrosystemic circle. The girls noted that embroidery allowed them to express their feelings about taboo subjects and protest in a subtle way. Embroidery has a long history of use as a means of conveying the plight and struggles of marginalized populations, particularly women [5]. It has been interpreted as a form of protest, for example, the silent protests of women who were hospitalized in mental institutions and who expressed themselves through embroidery [22,108]. Embroidery has also been used to protest against hardship in marginalized communities and has played a role in more extroverted protests such as the suffragist movement. Throughout history, the ability of embroidery to convey powerful messages of protest has been a hallmark of the craft [109,110].

## 5. Limitations and Conclusions

The present study has several limitations that need to be acknowledged, including the relatively small sample size and the potential influence of the prior acquaintance between the first author, who moderated the focus groups and conducted the interviews with the participants. Having said that, in order to promote maximum freedom of expression, the researcher avoided moderating the embroidery art therapy sessions, and the art therapists were replaced to minimize bias in Group A. Group B was monitored by two art therapists from the beginning. These arrangements were designed to keep the discourse open and enable unbiased results. In addition, in order to address potential biases related to the position of the first author, the Youth Participatory Action Research (YPAR) approach was used, which involved methodological triangulation and collaboration between the two researchers during the analysis process. In addition, the study featured a small sample of Israeli adolescents, and it is possible that different conclusions would have been reached if the participants were from a different cultural background. Therefore, the generalizability of our findings may be limited. Nonetheless, given the scarcity of research on this subject, this study provides valuable insights that can inform future studies. Thirdly, it is important to note that this study focused solely on the perspectives of adolescent girls who chose embroidery as part of their treatment, which may have led to a reduction in negative or objectionable references. However, our study aimed to map the therapeutic elements that arise from working with embroidery, including both complicating factors and those that provide relief.

In conclusion, this study demonstrates how embroidery can facilitate a dialogue regarding the self, groups, and cultural identities as part of the recovery process of adolescent girls who have experienced a mental crisis. The descriptions of the participants’ experiences offer valuable insights into the therapeutic potential of embroidery and can inform the practice of creative art therapists working with this population.

## 6. Practical Implications for Research and Clinical Work

Adolescent girls’ perspectives on their post-psychiatric hospitalization therapeutic experiences of embroidery can offer valuable insights and have practical implications for art therapy. One example that art therapists can incorporate into their therapeutic work is the importance of balancing structured and non-structured techniques. Patterns can provide adolescents, particularly those who have recently been hospitalized, with a sense of structure and order amid the chaos, while playfulness and flexibility can help them develop a more open and less rigid point of view. Embroidery is an optimal medium for achieving this balance due to the range it offers—from use of pattens to freestyle needlework. Additionally, therapists should consider the calming effect of embroidery and integrate it into their practice, especially when working with clients who struggle with emotional dysregulation.

Another implication of this study, which emerged from the exceptional versus conventional theme, is the need to tailor the type of therapy, the therapeutic language, and the therapeutic materials to the target audience since, as the participants noted, embroidery does not suit everyone. Furthermore, the recovery model (which focuses on strengths and healthy aspects of the self) is recommended as a theoretical and practical framework for adolescents after hospitalization. Time is also an important factor, and therapists should consider the period and context in which they engage in therapeutic work, particularly for adolescents who are building their identity with an eye towards the future. Taking that into account, it seems that therapists could even encourage an open conversation about this issue.

Finally, art therapists can use embroidery as a means of encouraging clients to express their feelings about taboo themes or topics. This approach can help patients work through latent issues and promote greater self-awareness and expression.

## Figures and Tables

**Figure 1 children-10-01084-f001:**
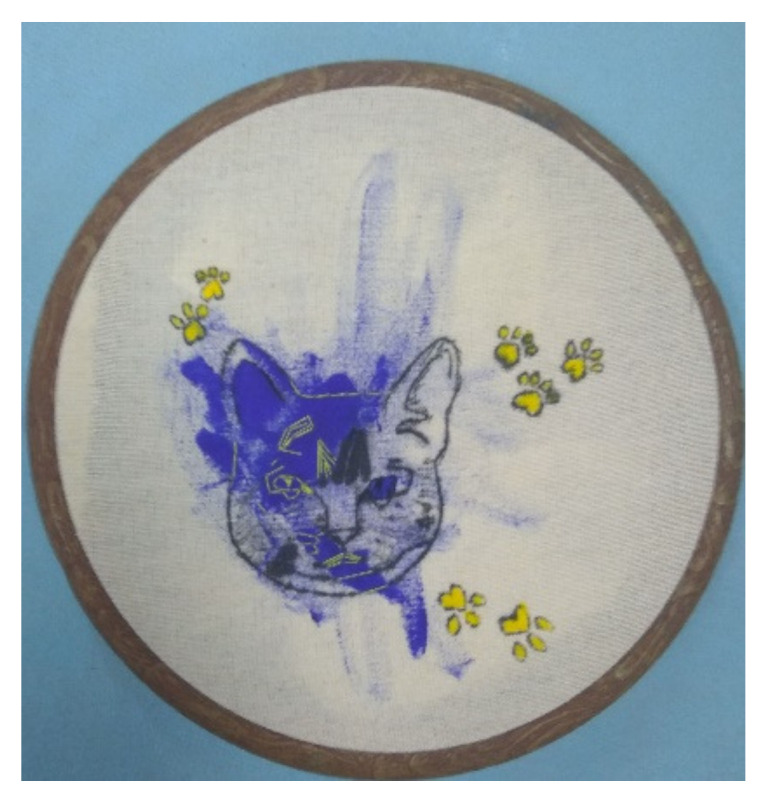
An embroidery with a cat figure combined with blue acrylic paint.

**Figure 2 children-10-01084-f002:**
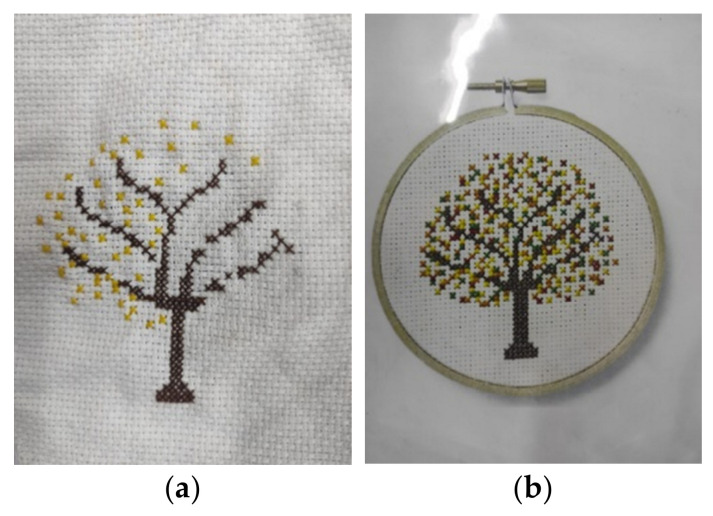
(**a**) A cross-stitch embroidery of a tree. (**b**) The cross-stitched embroidery pattern that informed the design of (**a**).

**Figure 3 children-10-01084-f003:**
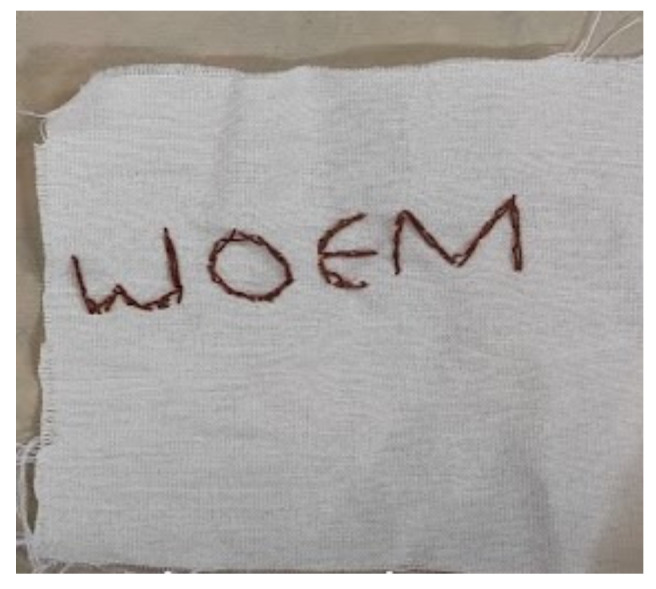
An embroidery of the word ‘meow’ that was embroidered upside down.

**Figure 4 children-10-01084-f004:**
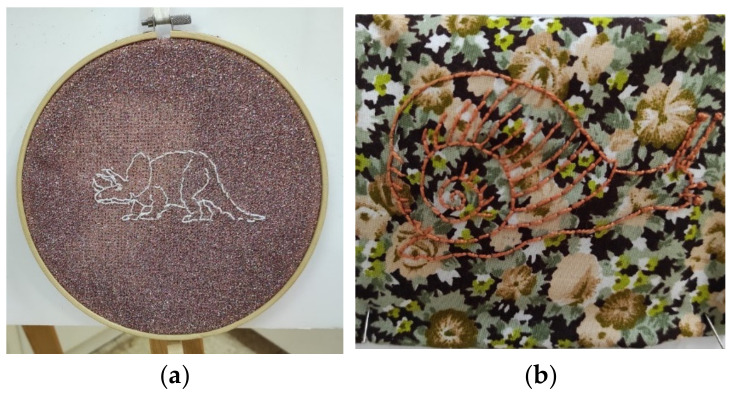
(**a**) A rhinoceros embroidered on a shimmering fabric. (**b**) A snail figure embroidered on a flowery background.

**Figure 5 children-10-01084-f005:**
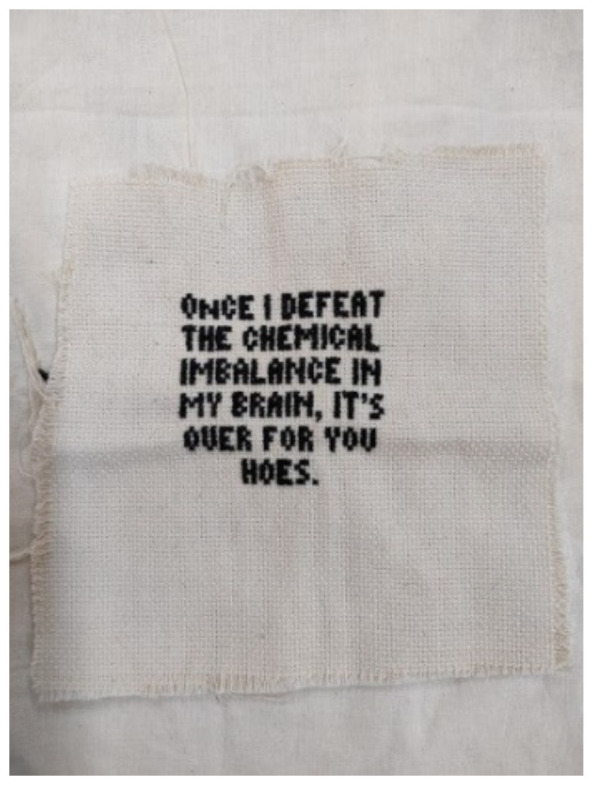
An embroidered piece with the text: “Once I defeat the chemical imbalance in my brain, it’s over for you hoes”.

**Figure 6 children-10-01084-f006:**
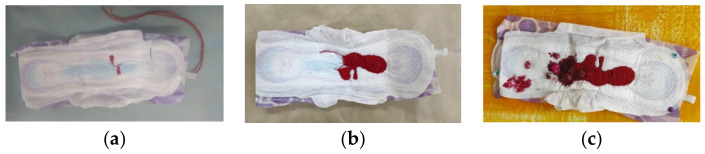
The process of embroidery depicting menstruation, embroidered on a menstrual pad. (**a**) The beginning of the embroidery. (**b**) The continued embroidery. (**c**) The finished product combined with beads.

**Table 1 children-10-01084-t001:** The distribution of three bioecological circles among the study’s five themes.

Theme	Bioecological Circle	Prevalence
Control versus release/freedom	Micro	Most of the Participants
	Meso	Most of the Participants
	Macro	Most of the Participants
Calmness	Micro	Most of the Participants
	Meso	Half of the Participants
	Macro	Half of the Participants
Exceptional vs. conventional	Micro	Most of the Participants
	Meso	Most of the Participants
	Macro	Half of the Participants
Stitch through time	Micro	Most of the Participants
	Meso	Half of the Participants
	Macro	Half of the Participants
Overt-latent	Micro	Half of the Participants
	Meso	Half of the Participants
	Macro	Few of the Participants

Legend: “Few”—2–4 participants; “Half”—3–7 participants; “Most”—8–10 participants.

## Data Availability

Data available on request due to restrictions, e.g., privacy or ethical. The data presented in this study are available on request from the corresponding author. The data are not publicly available due to privacy laws and ethical considerations.

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
