# Peer review of "The Therapeutic Aspects of Embroidery in Art Therapy from the Perspective of Adolescent Girls in a Post-Hospitalization Boarding School"

_children, 2023, doi:10.3390/children10061084_

Round 1

Reviewer 1 Report

The analysis is quite complex to demonstrate the therapeutic process of embroidery. I humbly appreciate your effort. However, it is unclear what types of mental disorders the girls were diagnosed with as it was mentioned 'a wide range of mental disorders.  There should be some discussion points with reference to variation in therapeutic experiences in relate to expereince of mental disorders 

It is quite complex and I do not feel that I cannot maintain focus and understanding of how the authors are connecting the dots. 

Author Response

 Firstly, we would like to thank you for your comments. We appreciate your feedback and would like to address your concerns. 

  1. Regarding your comment on the unclearness of what types of mental disorders the girls were diagnosed with, we added a clarification in the Participants section. The participants were students at a post-psychiatric hospitalization boarding school known for its treatment of a wide range of mental disorders, including depression, anxiety, eating disorders, personality disorders, accentuation of personality traits, PTSD and adjustment disorders, suicidal ideation, self-harm, ADHD, and learning disabilities.

Please see the attached paragraph after the revision: 

"2.2 Participants

13 Israeli adolescent girls aged 15-18 (M=17.42, SD=1.164) participated in this study. They were students in a post-psychiatric hospitalization boarding-school that is known for its treatment of a wide range of mental disorders that includes: Depression, anxiety, eating disorder, personality disorders, accentuation of personality traits, PTSD and adjustment disorder, suicide ideation, self-harm , ADHD and learning disabilities. 

All participants had attended for a period of at least 12 months to 2 years embroidery groups held at the boarding-school in an open studio setting during 2020 – 2022. Participants were selected using criterion sampling [‎87]."

    We apologize for any confusion caused by the initial lack of clarity regarding the specific types of mental disorders. We have now provided a comprehensive list to address this concern. 

  1. Regarding your comment that there should be some discussion points with reference to variation in therapeutic experiences in relation to experience of mental disorders. We have addressed the issue in the attached paragraph, explaining that this research is based on the open studio approach in art therapy. The open studio approach is a dynamic psychotherapeutic approach that considers a variety of mental difficulties as a mental state, as defined by the United Nations Human Rights Council. While this approach does not specifically address each disorder individually, it offers a compassionate and inclusive solution that emphasizes the recovery process for the range of difficulties and challenges teenagers face after hospitalization.

Please see the attached paragraph after the revision, also references were added to clarify - in the section: Out-of-Home Art Therapy that Addresses Mental Health Challenges in Adolescents  ( lines 134-142)

"Art therapy is administered to vulnerable adolescents in boarding schools who experience interpersonal and intrapersonal communication challenges [‎74-‎76]. It provides an alternative means of expression that circumvents speech, which can often be perceived as threatening [‎77].  In line with the evolving understanding of mental health, there has been a significant shift in terminology at a global level. Today, the United Nations Human Rights Council has replaced the term 'mental illness' with 'mental health conditions' (MHC) [‎78]. This new terminology represents a transition from a medical approach to one focused on wellbeing and a shift in emphasis from pathology to recovery [‎79].

The open studio approach, in its essence, aligns with this progressive shift. It acknowledges the importance of holistic well-being and recovery rather than solely focusing on the medical aspect"

For the sake of clarity we added the additional relevant articles that were mentioned in the above paragraph:

[78]Kelemen, L.J.; Shamri-Zeevi, L. Art Therapy Open Studio and Teen Identity Development: Helping Adolescents Recover from Mental Health Conditions. Children 2022, 9, 1029, org/10.3390/children9071029

[79]Tuaf, H.; Orkibi, H. Community-Based Rehabilitation Programme for Adolescents with Mental Health Conditions in Israel: A Qualitative Study Protocol. BMJ Open. 2019, 9, e032809, doi: 10.1136/bmjopen-2019-032809

3.Thank you for referring us to issue of the unclarity of the English in the article. However, The article has already undergone professional linguistic editing, yet, if you're still concerned about this issue we kindly request that you provide more specific details regarding the areas where further clarification is needed. 

We sincerely appreciate your time and effort in reviewing the article and offering your valuable comments. Your insights have been highly valuable to us. 

Reviewer 2 Report

Dear Authors,

I appreciate your labor in conducting and writing this manuscript related to embroidery art therapy. The embroidery art therapy, the importance of the adolescence period, and using the art therapies among adolescents are explained well. The method of the study is precise. I have some minor recommendations for the manuscript. 

- Please explain the sample, for example, which mental disorders they had or the inclusion criteria of the sample could be added. 

- Please add the legal guardian or the parents' informed consent approval in the ethical consideration since this population is vulnerable. 

I wish you success in your work. 

Author Response

Thank you for your valuable comments.

  1. In response to your request for a specification of the mental disorders experienced by the girls in our study, we have provided a detailed description in the attached paragraph below ( lines 199-201 in the Article). Here's the quote of the revised text:

"2.2 Participants

13 Israeli adolescent girls aged 15-18 (M=17.42, SD=1.164) participated in this study. They were students in a post-psychiatric hospitalization boarding-school that is known for its treatment of a wide range of mental disorders that includes: Depression, anxiety, eating disorder, personality disorders, accentuation of personality traits, PTSD and adjustment disorder, suicide ideation, self-harm, ADHD and learning disabilities. 

All participants had attended for a period of at least 12 months to 2 years embroidery groups held at the boarding-school in an open studio setting during 2020 – 2022. Participants were selected using criterion sampling [‎87]."

  1. Thank you for your comment regarding the inclusion of legal guardian confirmation or informed consent of the parents in the ethical considerations. We received your comment and tried to elaborate about this issue. We hereby attached the paragraph below for further clarification ( lines 211-214 in the Article):

"After receiving ethical approval from the Ministry of Labor, Welfare and Social Services and the Ethics Committee of the Faculty of Social Welfare and Health Sciences at the University of Haifa, we invited the adolescents who participated in the embroidery sessions in the open studio to take part in the YPAR, which focused on the experiences of embroidery in reference to the embroidered pieces. In order to ensure the involvement of minors and address the vulnerability of this population, we obtained explicit consent from the parents or legal guardians for the participation of all 13 girls. This consent was obtained in addition to the ethical approval granted."

We appreciate your feedback, and we want to assure you that we have taken the necessary steps to prioritize the well-being and informed consent of the participants, especially considering the involvement of minors. If you have any further questions or require additional information, please let us know.

Thank you for your time and effort in reviewing the article and offering your valuable comments. 
